# Ecological patterns in the Porto-Novo Lagoon (Benin, West Africa): A review with implications for SDG 6.3.2 and EU-WFD readiness toward ecological status classification

Metogbe Belfrid Djihouessi[1,2*], Gildas Djidohokpin[3,4], Romaric Christian Marc Hekpazo[1,2,5], Akilou Amadou Socohou[1,2,5], Lewis Zounon[3], Zacharie Sohou[3,6], Abou Youssouf[7], Martin Pépin Aina[2,5]

**1** Centre d'Expertise et de Recherche en Eau, Sol et Environnement, Abomey Calavi, Benin, **2** Laboratoire des Sciences et Techniques de l'Eau et de l'Environnement, Université d'Abomey-Calavi, Abomey Calavi, Benin, **3** Laboratoire de Recherche sur les Zones Humides, Département de Zoologie, Université d'Abomey Calavi, Abomey Calavi, Benin, **4** Département de Gestion des Ressources Naturelles, Faculté des Sciences de l'Environnement, Université de N'Zérékoré, N'Zérékoré, République de Guinée, **5** Centre d'Excellence Africain pour l'Eau et l'Assainissement, Institut National de l'Eau, Université Abomey-Calavi, Abomey Calavi, Benin, **6** Institut de Recherche Halieutique et Océanologique du Bénin, Cotonou, Benin, **7** Laboratoire d'Ecologie et de Management des Ecosystèmes Aquatiques (LEMEA), Faculté des Sciences et Techniques, Université d'Abomey-Calavi, Abomey-Calavi, Benin

* d.belfrid@aims.edu.gh

## Abstract

Urban tropical lagoons provide vital services yet face nutrient loading, macrophyte blooms, and episodic hypoxia. This scoping review compiles four decades of hydro-morphological, physico-chemical, and biotic evidence for Benin's Porto-Novo Lagoon and assesses readiness for UNEP SDG 6.3.2 Level 1 reporting and a EU Water Framework Directive (WFD)-oriented exploratory gap analysis rather than an opera-tional assessment. The lagoon is strongly seasonal, shaped by Ouémé inflows and marine intrusions; eutrophication symptoms and floating macrophytes have intensi-fied since the 1980s. Core SDG 6.3.2 Level 1 parameter-group proxies (pH, DO, EC, $PO_4$-P) are moderately covered, but nitrogen evidence is scarce and time-continuous station series are rare. A completeness audit indicates about 60% coverage for SDG-related variables and lower maturity for WFD biological elements, meaning that formal SDG compliance-based classification ("good and not good") and any Eco-logical Quality Ratios based WFD ecological status classification cannot be derived from the compiled literature. Near-term priorities are therefore framed as readiness building, focused on strengthening consistent in situ series for the SDG core groups; context-adapted complementary sources may support gap filling and interpretation but do not substitute Level 1 requirements. For WFD, the evidence base highlights prerequisites for future work, including transitional water-body delimitation and type-specific reference conditions, standardised Biological Quality Elements monitor-ing, and priority-substance surveillance. Comparison with Ébrié, Lekki, and Sakumo

**Data availability statement:** All relevant data are within the manuscript.

**Funding:** The author(s) received no specific funding for this work.

**Competing interests:** The authors have declared that no competing interests exist.

II underscores the value of governance and monitoring designs that avoid data fragmentation.

---

## 1. Introduction

### 1.1. Global context of urban tropical lagoons

Tropical coastal lagoons in urban areas play a major role in the ecosystem. They provide essential habitats and numerous ecosystem services (nurseries for fish and crustaceans, feeding grounds for migratory birds, protection against storm surges, etc.) [1–3]. These environments are highly productive thanks to the recycling of nutrients by their biota, which supports important local fisheries. However, urban lagoons are subject to multiple anthropogenic and environmental pressures. Some are located at the outlet of large basins receiving significant amounts of agricultural effluent. In addition, rapid urbanisation around these water bodies is accompanied by the discharge of untreated wastewater and solid waste. These situations lead to high levels of organic and nutrient pollution, resulting in the proliferation of invasive macrophytes (*Typha domingensis*, *Eichhornia crassipes*) and recurrent episodes of nocturnal hypoxia [4–6] Cliquez ou appuyez ici pour entrer du texte.Cliquez ou appuyez ici pour entrer du texte.. These situations are exacerbated by a lack of monitoring infrastructure: a review of 42 African studies [7] showed that more than 70% used Water-Quality Indices that were sometimes unsuitable and difficult to compare between basins. Even where monitoring stations exist, the lack of long time series makes it impossible to distinguish between natural trends and interannual fluctuations [8,9].

Despite these challenges, the majority of developing countries still do not have national standards for holistically classifying the ecological status of water bodies, which would enable the formulation of clear objectives for the sustainable management of urban lagoons. Water agencies most often rely on a few locally unvalidated physico-chemical thresholds and biological indicators, inherited from bilateral projects or drawn from assessment frameworks applied in developed countries [10,11] Cliquez ou appuyez ici pour entrer du texte. However, the indicators drawn from these assessment frameworks developed in temperate regions (e.g., the European Union Water Framework Directive or the Canadian Environmental Quality Guidelines) are based on 'pristine' reference conditions that are rarely applicable in the tropics. This is particularly problematic given that one of the major challenges facing tropical urban lagoons is the lack of historical reference data; insufficient monitoring thus complicates the establishment of local ecological quality standards. Many developing countries lack the capacity to regularly monitor water quality [12–14]. Globally, less than 3% of the data used to assess the water quality indicator according to UNEP GEMS/Water (rated SDG 6.3.2) comes from the poorest half of countries, illustrating the lack of monitoring of tropical waters [15]. This normative deficit explains the difficulty of establishing a rigorous and official ecological classification for lagoons in the West African sub-region, which constitutes an obstacle to the sustainable management of urban lagoons.

## 1.2. International frameworks for the ecological classification of surface waters

At the international level, several frameworks aim to objectify and compare the status of waters based on common measurement and decision-making rules. Two complementary frameworks dominate: either a 'minimalist-operational' approach focused on physical and chemical variables to provide a summary diagnosis in contexts where data is limited, or an 'integrative-typological' approach that aggregates biological indicators, supporting conditions and hydromorphological determinants to qualify ecological status.

At the global level, the UNEP GEMS/Water programme is piloting the implementation of SDG indicator 6.3.2 'Good environmental water quality' [16]. This is based on five mandatory parameters (pH, DO, EC, $NH_3$-N/$NO_2$-N, $PO_4$-P) classified according to four thresholds (I to IV). A water body is considered 'good' if ≥ 80% of the measurements collected over five years fall within classes I or II. The result is therefore binary (good/not good), but provides a minimum harmonised basis for countries that do not have biological reference standards [17]. The 2017–2022 reporting wave shows that only 22% of African watercourses and water bodies achieved this target, compared to 62% in Europe and 48% in Latin America [18].

The European Union#39;s Water Framework Directive (WFD) establishes a five-state typology (very good, good, moderate, poor, bad) based on: (i) Biological Quality Elements (BQE) (phytoplankton, macrophytes, benthic macroinvertebrates, fish); (ii) supporting physico-chemical elements (nutrients, oxygen, transparency, specific pollutants); and (iii) hydromorphology, including hydraulic regime and bank integrity [9]. The assessment is based on the 'one-out, all-out' principle: the most degraded result determines the final score and conditions the restoration objectives. Although designed for temperate contexts, the WFD has inspired several emerging countries (e.g., Brazil, South Africa) to establish their own ecological status grids [19,20].

These two frameworks therefore have different levels of requirement (see Supporting Information S1 File, Section S1 for details). The lighter SDG 6.3.2 tool allows for an initial summary diagnosis, which is useful for countries with limited resources, but it remains blind to hydromorphological and ecotoxicological alterations. The WFD, which is very detailed but cumbersome to implement, requires typology, reference conditions and Ecological Quality Ratios (EQR)-based biological assessment. The purpose of this study is to examine the extent to which the data already available for a typical West African lagoon, the Porto-Novo Lagoon, support (i) SDG 6.3.2 reporting and (ii) a WFD-oriented gap/readiness analysis identifying what is missing to enable any future WFD-inspired ecological status classification for transitional waters in West Africa

## 1.3. Porto-Novo Lagoon: Local issues

The Porto-Novo lagoon (Fig 2), located in south-eastern Benin, is a prime example of a tropical lagoon subject to significant anthropogenic pressures. Located between parallels 6°25' and 6°30' North and meridians 2°30' and 2°38' East, it is part of the fluvial-lagoon complex of the lower Ouémé River delta, from which it receives fresh water from the north-west [21,22]. It is connected to Lake Nokoué in the south-west via the Totchè Channel, and to Lagos Lagoon in Nigeria in the east via Badagry Creek. The Porto-Novo lagoon is doubly connected to the Atlantic Ocean via these two ecosystems. It is located in the sub-equatorial climate zone, characterised by two rainy seasons (April to mid-July and mid-September to mid-November) with an average annual rainfall of around 1,200 mm and temperatures ranging from 26 to 33°C [23].

More than 500,000 inhabitants depend directly on this lagoon for their livelihoods, including small- scale fishing, transport and coastal agriculture. Local communities (e.g., the Tofinou, Wémènou and Xuéda ethnic groups) have developed a culture closely linked to the lagoon, particularly through traditional fishing [21,24]. Over the last few decades, numerous scientific studies have documented a gradual deterioration in the ecological status of the Porto-Novo lagoon, potentially caused by various anthropogenic pressure factors: (i) traditional fishing practices such as acadja, (ii) untreated domestic and agricultural effluent discharges, and (iii) erosion of the banks and watershed. Each of these factors contributes to altering the water quality and aquatic habitats of the lagoon system.

## 2. Methodology

The approach used in this study is based on the scoping review model, which is appropriate when the literature is heterogeneous, and no single meta-analysis can be performed [25,26]. It follows the PRISMA-ScR approach, a reporting guideline for scoping review [27,28]. PRISMA-ScR was used here to improve transparency and reproducibility of reporting, by explicitly documenting the information sources, search strategy, screening workflow, data extraction framework, and quality-control steps (Fig 1). The date of the last search was September 2025.

### 2.1. Documentary research approach

The literature review was conducted in two stages. The first stage involved compiling the main Porto-Novo corpus, while the second stage involved creating the West African comparative corpus. Structured bibliographic databases (Scopus, Web of Science, AJOL) and institutional repositories (HAL, IRD Horizon) were searched in French or English. In parallel, Google Scholar was used as a web-based search engine to capture additional records not consistently indexed in bibliographic databases. The date of the last search was September 2025.

The search was implemented using database-adapted Boolean queries, built around a common core string: ("Porto-Novo" OR "Porto Novo" OR "Porto-Novo Lagoon" OR "Lagune de Porto-Novo") AND (water OR sediment OR nutrient OR phosphorus OR nitrogen OR "water quality" OR phytoplankton OR macrobenthos OR "benthic invertebrate*"

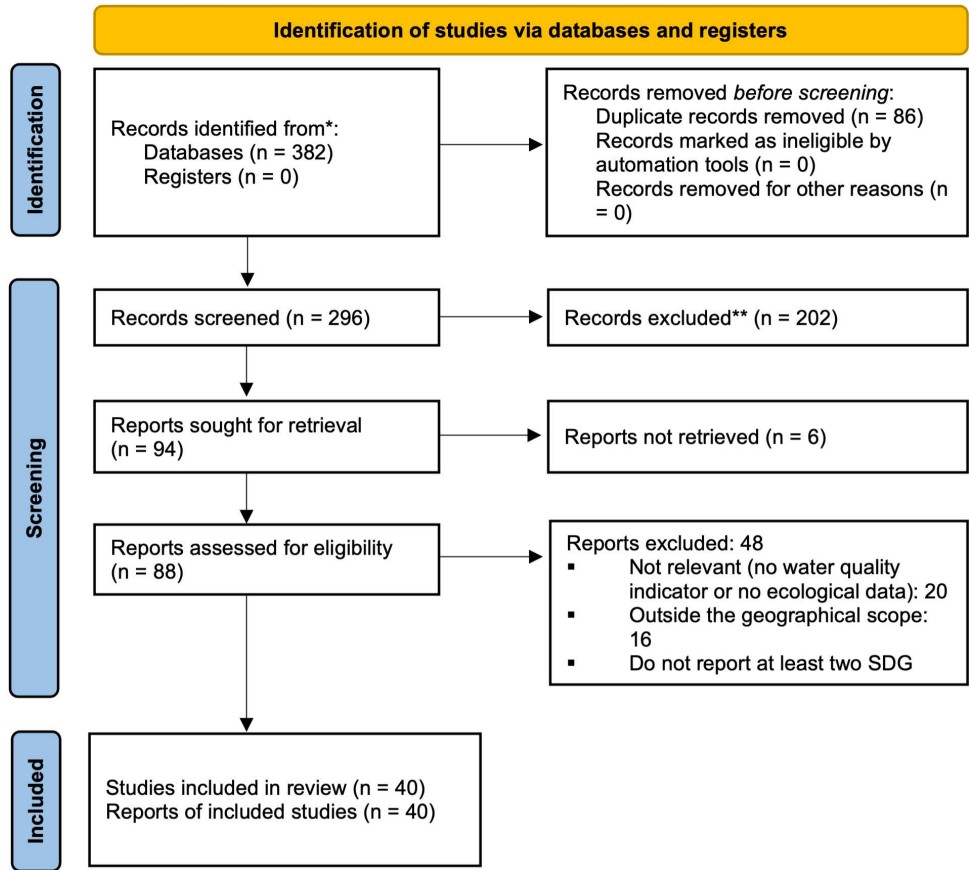

**Fig 1. PRISMA 2020 flow diagram of study selection.**

OR fish* OR macrophyte* OR "aquatic vegetation" OR hydromorpholog*). The query was applied to the TITLE-ABSTRACT-KEYWORDS field in Scopus; to the TOPIC field in Web of Science; and to the available search fields (typically title/abstract/keywords or full text) in AJOL and institutional repositories. For Google Scholar, the query was implemented as a free-text search using the same core concepts and synonyms. The limits were set to 1970–2024. The references were imported into Mendeley 2.94, deduplicated and then screened for titles, abstracts and full texts according to the inclusion/exclusion criteria detailed in section 2.3. Screening was conducted in two steps: first, the titles and abstracts were screened, and then the full texts were assessed, in line with the PRISMA flow diagram (Fig 1).

For each analogous lagoon of Porto Novo lagoon: Lagos/Lekki, Ébrié, Sakumo II), the query was expanded by replacing the geographical term; only documents published since 2010 and reporting at least two indicators relevant to the SDG 6.3.2 or WFD-oriented evidence mapping (physico-chemistry, BQE, hydromorphology) were retained. When several studies met these criteria, the most recent and comprehensive ones were given priority, resulting in a maximum of six references per lagoon.

## 2.2. Processing method

In accordance with the PRISMA-ScR recommendations Tricco et al., 2018 [27] and the Arksey et al., 2005 [25] scoping study framework, bibliographic metadata (DOI, year, document type) were imported automatically, and then a 22-field extraction mask was filled in manually (see Supporting Information S2 Checklist). A controlled vocabulary was applied for variables and matrices to reduce ambiguity across heterogeneous sources. When certain information was missing from a document, the field was coded "not reported" rather than left blank, in order to maintain a consistent table format while recognising the heterogeneity of the sources.

Quality control focused on the internal consistency of the charted dataset. Extracted values were checked for unit consistency and harmonised where possible; otherwise, the original units were retained and flagged. Duplicate reports of the same dataset were cross-checked to avoid double counting and the most complete source was retained as the primary record, with linked references noted. Internal consistency checks were performed to identify obvious transcription errors, such as mismatched units, impossible ranges, or duplicate site-season entries. Finally, a completeness audit was performed to quantify evidence coverage by thematic blocks aligned with the main evidence domains (physico-chemistry, biological quality elements and hydromorphological descriptors) and by framework (SDG 6.3.2 and WFD; Section 2.4).

## 2.3. Inclusion and exclusion criteria

The following were included: (i) studies containing original quantitative data collected in situ in one of the four lagoons; (ii) institutional reports explicitly describing measurement protocols; (iii) theses and dissertations defended at accredited institutions. The following were excluded: (i) records not referring to the target water bodies; (ii) studies without in situ quantitative measurements; (iii) purely modelling or opinion papers not supported by primary measurements; (iv) articles with unclear or non-reproducible methodology; and (v) duplicate reporting of the same dataset.

## 2.4. Analysis of publications in relation to SDG 6.3.2 and WFD frameworks

Extracted variables from literature review were mapped against SDG 6.3.2 Level 1 core parameter groups (operationalised here with pH, DO, EC, $NH_3$-N/$NO_2$-N, $PO_4$-P) and against WFD quality-element evidence needs (BQEs, supporting physico-chemical elements, and hydromorphological descriptors).

For SDG indicator 6.3.2, a clear distinction is made between formal UNEP Level 1 reporting (classification of representative water bodies as "good/not good" based on in situ monitoring of five core parameter groups and national target values) and the context-adapted evidence mapping performed in this literature-based review (see Supporting Information S1 File, Section S3). Availability of the Level 1 core parameter groups (oxygen, salinity, nitrogen, phosphorus and acidification) is assessed using the variables most consistently reported in the literature (DO, EC, pH, $NH_3$-N/$NO_2$-N, $PO_4$-P).

The threshold of ≥10 observations over five years is used as an internal readiness criterion (not an UNEP requirement); given heterogeneous sources and the absence of consistently applicable national targets, an official Level 1 "good/not good" classification is not derived from the compiled evidence.

For the WFD component, an ordinal data-readiness (degree of completeness) score (0–4) was used to benchmark the availability and structuring of evidence for each quality element against key WFD classification prerequisites (see Supporting Information S1 File, Section S2). Scores were defined as follows: 0/4 = absent; 1/4 = qualitative or site-specific evidence only; 2/4 = inventories exist but without consistent time series and/or without reference context; 3/4 = repeated quantitative series exist but reference conditions are not defined; and 4/4 = prerequisites for a traceable WFD classification are met. This scoring is heuristic and was not intended as a validated metric; it is not an EQR and does not produce WFD status classes, because typology, reference conditions and class boundaries are not established for the Porto-Novo Lagoon. For each quality element, the assigned score reflects the best available evidence meeting these criteria across the compiled sources; duplicate reporting of the same dataset was cross-checked to avoid double counting, retaining the most complete record as primary. Aggregated "degree of incompleteness" summaries were computed with equal weighting across parameters within each framework block, using: *Incompleteness (%) = 100 × [Σ(max score − observed score) / Σ(max score)]*; where *max score* is the maximum possible score per element, and *observed score* is the assigned score. Accordingly, the scoring should be interpreted as indicative gap mapping rather than a quantitative measure of distance to compliance or ecological status.

## 3. Results

### 3.1. Hydromorphology

#### 3.1.1. Seasonal hydrological dynamics.
The state of knowledge on the hydrological cycle of the Porto-Novo lagoon is relatively well documented. Studies describe a highly seasonal system governed by river inflows, marine intrusions, precipitation and marked evaporation during the dry season. The annual regime is divided into two contrasting phases: the high water period (August-October), marked by massive flooding from the Ouémé River and increased urban runoff, and the low water period (December-April), characterised by low river levels, high potential evaporation and sustained saline intrusion from Lake Nokoué and Badagry Creek [29,30]. The inter-seasons (May-July and November-December) provide the transition between these extremes. It should be noted that local rainfall has only a secondary influence compared to the descent of water from the Ouémé River basin.

The Ouémé River plays a major role in the hydrology of the lagoon, carrying almost all of the freshwater and suspended matter during the high-water period. Peak flows reaching 1100 m³ s⁻¹ at Bonou (Fig 2) desalinate the lagoon. The highest velocities are concentrated along the north-south axis (Totchè-Djassin). During this period, the waters of the Porto-Novo Lagoon flow through the Totché canal to Lake Nokoué [31].

During low water periods, marine intrusion into the lagoon occurs via Lake Nokoué and the Totché channel, with surface water currents exceeding 2 m.s⁻¹ [32]. Combined with the water deficit, i.e., the almost zero inflow from the Ouémé River and sustained evapotranspiration of around 6 mm.d⁻¹, this phenomenon causes near-stagnation and hypersalinity. As a result, salinity in March-April can reach 19‰, and may even exceed 28‰ in confined areas [33].

#### 3.1.2. Morphological characteristics.
The lagoon has an elongated west-east shape, varying in length from 6 km to 7 km, with a width of between 2 and 4 km [22,34]. Its surface area fluctuates greatly with the hydrological regime: approximately 20 km² at low water and 30 or 35 km² during flood periods [32], representing a seasonal amplitude of nearly 50%, resulting in lateral connectivity with the wetlands to the west and north (Fig 3). Recent measurements at low water have specified average depths of 0.75 ± 0.15 m in the west, 1.10 ± 0.20 m in the centre (at Djassin) and up to 2.00 ± 0.60 m in the east (at Agbokou), suggesting a west-east increasing depth profile.

#### 3.1.3. Sediment properties and quality.
Historically, the sediments of the Porto-Novo lagoon have been described as a sandy-clay lithofacies composed of approximately 20% sand, 14% silt and 66% intermediate facies, with a clay fraction

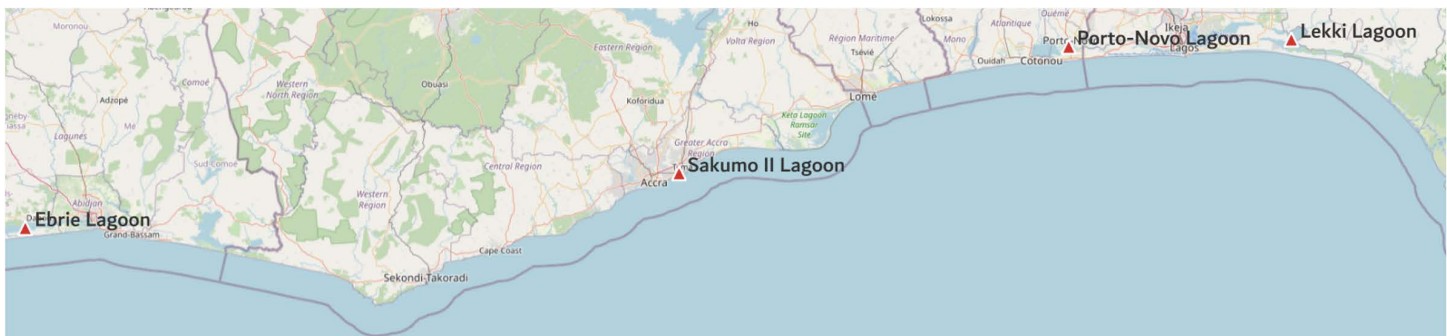

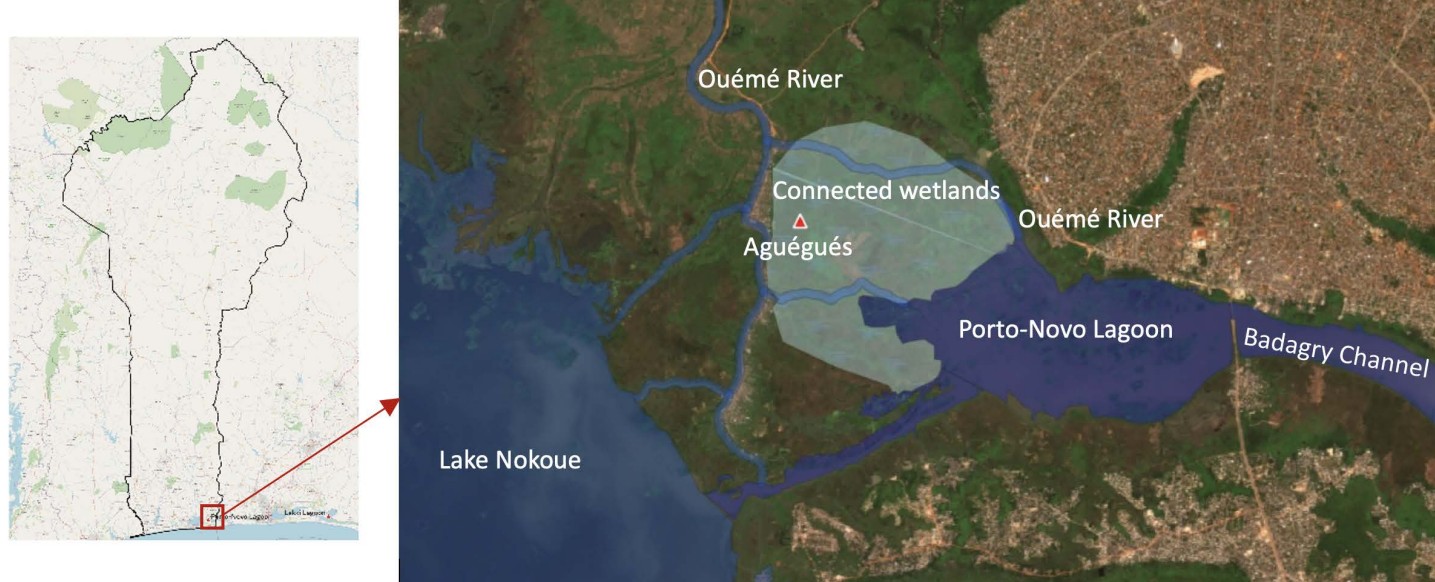

**Fig 2. Porto-Novo Lagoon (Benin) and the West African comparison lagoons (Ébrié, Lagos/Lekki, Sakumo II): location map based on Sentinel-2 Level-2A imagery (20 December 2024; Copernicus Open Access Hub, Copernicus Programme; open data policy) processed in QGIS.**

dominated by smectites and kaolinite [35]. However, no recent, comprehensive survey of sediment composition has been published. Subsequent work, has nevertheless shown that organic matter of plant origin is abundant in the sediments, suggesting significant inputs linked to the accumulation of debris from senescent water hyacinths and water lilies, particularly after episodes of increased salinity [36] (Mama, 2010).

The average pH of sediments sampled in the lagoon is around 6.9. The lowest values of organic matter and carbonate are generally observed in the north, while the highest values are reached in the centre [33,37]. Studies strongly suggest that sediments in the Porto-Novo Lagoon act as traps for phosphorus and nitrogen. The fraction of phosphorus bound to organic matter is the most representative in sediments, reaching an average of 10.6 mg-P/g, or 68.76% of the average value of total phosphorus contained in sediments.

Multiple studies have revealed significant pollution of sediments by trace metals: Fe levels vary between 1.3 and 25.7 mg/g, Mn between 0.5 and 5.4 mg/g, Cu between 2.1 and 4.6 mg/g and Zn between 0.45 and 0.70 mg/g in surface sediments. The spatial heterogeneity of concentrations can be explained by the variable texture of the substrates and the strong currents in the Totchè Channel, which resuspend and redistribute fine particles, as well as by the burrowing of benthic invertebrates [32].

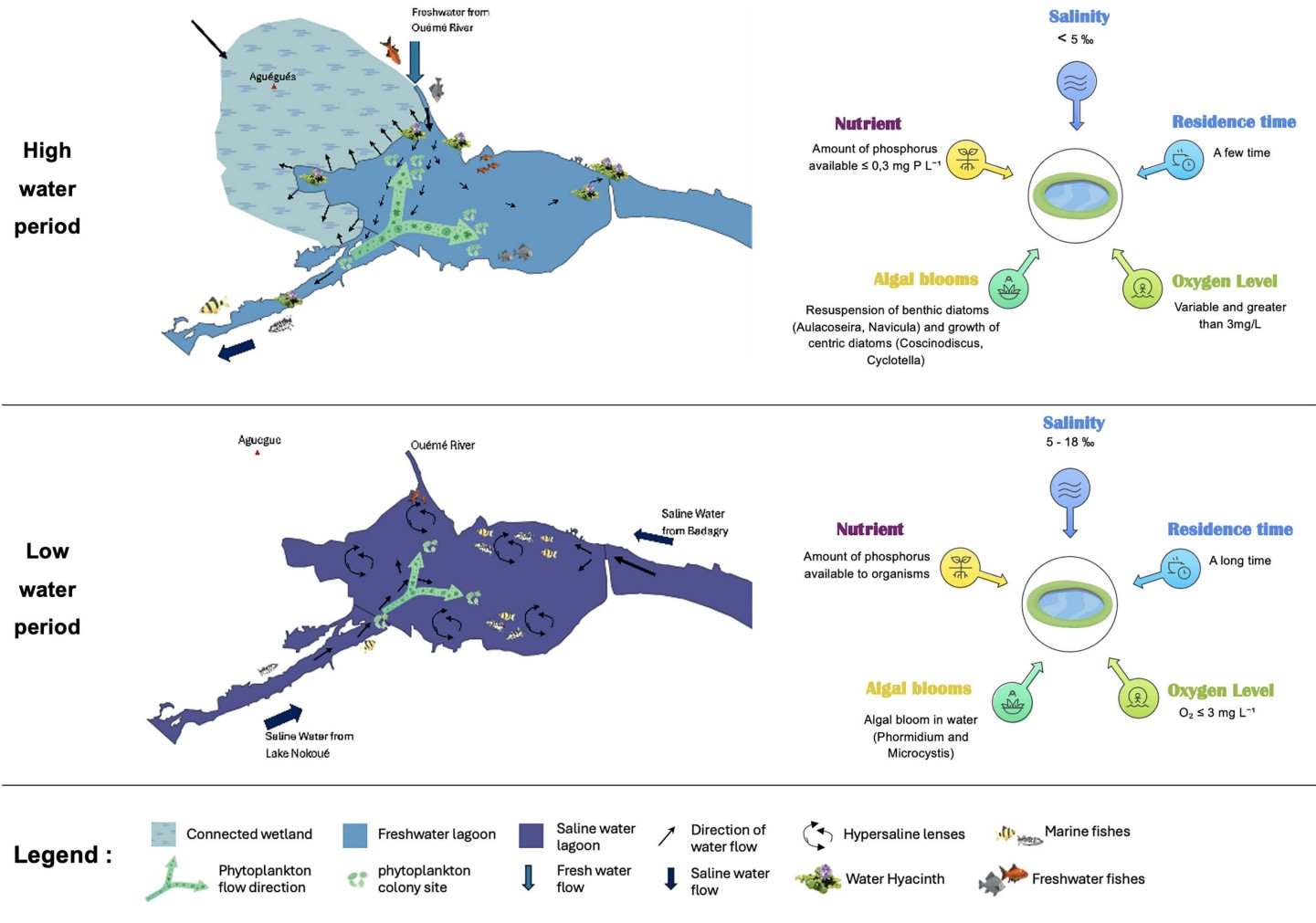

**Fig 3. Framework for describing seasonal ecological relationships in the Porto-Novo lagoon.**

### 3.2. Physicochemical water quality

**3.2.1. Historical trends.** The first studies on water quality between 1978 and 1980 described slightly brackish water that was well oxygenated (5−7 mg/L) and low in minerals, indicating an ecosystem that was still moderately disturbed [35]. Three decades later, the diagnosis has changed. Campaigns carried out between 2014 and 2016 showed a decrease in dissolved oxygen to less than 3 mg/L in all seasons and high average turbidity (86 ± 25 NTU) outside the flood period [32,38]. Mama et al. (2011) recorded average orthophosphate and total phosphorus levels of between 0.02–0.08 mg/L and 0.30 and 6.05 mg/L, respectively, and a COD varying between 100 and 142 mgO$_2$/L, indicating that the system has entered a state of hyper-eutrophication. At the same time, the pH range of 5.2 to 5.9 measured in 2020, well below the ranges of 7.2 to 7.6 measured in 1980 and 5.80 to 7.60 measured in 2011, suggests progressive acidification attributable to organic inputs [33,34,39].

**3.2.2. Spatio-temporal variability and major contaminants.** During periods of flood, dominated by freshwater inflows from the Ouémé River, salinity falls below 5 ‰, turbidity exceeds 250 NTU, and nitrate levels are around 1.10 mg/L, orthophosphates are around 0.04–0.05 mg L$^{-1}$ and dissolved metals remain moderate (Cu: 0.001–0.006 mg/L; Zn:

0.06–0.13 mg/L [33,38]. At low water levels, the water column shows (i) an enrichment in nutrients, partly linked to the post-anoxic release of mobile phosphorus from sediments (up to 0.55 mg.g$^{-1}$) and (ii) an increase in dissolved metals (Cu: 0.01–0.06 mg/L; Pb: 0.10–0.12 mg/L; Hg: 0.0008–0.0028 mg/L) [33,37,38,40].

Spatially, the Totchè Channel area (south-west, connected to Lake Nokoué) has the highest concentrations of marine ions (chlorides, sodium) and dissolved metals, while the lateral areas of the water body have higher concentrations of reduced forms of nitrogen ($NO_3^-$, $NH_4^+$), with a negative $NH_4^+$/$O_2$ correlation indicative of reducing conditions [33,37,38].

## 3.3. Biota

Biological evidence is synthesised here primarily to describe ecological patterns and pressures; unless explicitly stated otherwise, qualitative observations (presence or absence, narrative descriptions) are treated as contextual information and not as assessment-ready inputs for policy-relevant evaluation.

### 3.3.1. Phytoplankton.
Early studies on the phytoplankton community in the Porto-Novo Lagoon (1978–1980) only reported the presence of the major phyla Chlorophyceae, Cyanophyceae and Bacillariophyceae, without providing a species inventory or abundance data. The community was described by the authors as compatible with a moderately disturbed brackish environment [34]. Until the 2010s, studies remained fragmentary, with qualitative indications of algal blooms but no concrete data on diversity, abundance and seasonal variability.

The first comprehensive analysis of phytoplankton was carried out during the 2015–2016 hydrological cycle and identified 170 species belonging to 71 genera, 29 families and 9 classes [41]. In 2017, a second assessment identified 160 species divided into 11 classes [42]. These two studies revealed the co-dominance of Diatomophyta, Chlorophyta and Cyanophyta ahead of Euglenophyta. Densities vary from $8.3 \times 10^8$ to $1.6 \times 10^{10}$ cells/L with a maximum of $3.8 \times 10^9$ cells/L during the short rainy season.

Spatio-temporal variability is organised according to an increasing gradient from the south-west (Totché Canal) towards the north-east and north-west. The different seasons are characterised by:

- Low water: salinity 5–18‰, long residence time, $O_2 \leq 3$ mg/L; opportunistic blooms of *Phormidium* and *Microcystis,* Euglenophyceae (bioindicators of organic loads) become dominant, but overall biomass is moderate due to a lack of bioavailable phosphorus (≤ 0.05 mg P /L) [37].

- High water: salinity < 5‰, abundant bioavailable phosphorus reaching up to 0.2 mg/L; resuspension of benthic diatoms (*Aulacoseira, Navicula*) and proliferation of centric diatoms (*Coscinodiscus, Cyclotella*); densities ≥ $1.5 \times 10^6$ cells/L in stirred areas, according to Akogbeto et al. (2018).

Thus, two levers for controlling the phytoplankton community have been identified for the Porto-Novo lagoon: (i) salinity modulates species composition, while (ii) bioavailable phosphorus drives biomass.

### 3.3.2. Aquatic invertebrates.
The information available on aquatic invertebrates in the Porto-Novo lagoon remains sparse and heterogeneous. It comes mainly from one-off campaigns and studies focusing on fauna associated with fish or sediments.

*3.3.2.1. Zooplankton:* Zooplankton compartment was the least documented component of the lagoon. The first surveys (1978–1980) revealed a small community (< 20 ind.m⁻²) dominated by: Foraminifera: *Jadammina polystoma* (fresh to slightly brackish waters), *Ammonia beccarii* (mesosaline waters); Ostracods: *Neomonoceratina sp*. and Copepods [34]. No subsequent studies have specified the abundance or seasonal variability of these organisms.

*3.3.2.2. Benthic macrofauna:* The first documented inventory of the benthic macrofauna of the Porto-Novo lagoon was carried out in 1978–1980. This inventory reported a sparse but taxonomically diverse fauna. For nearly three decades, no large-scale monitoring was carried out to supplement these isolated observations. It was not until the early 2010s that a series of systematic studies documented the diversity, abundance and seasonal dynamics of benthic macroinvertebrates

[43,44], suggesting an impoverished community dominated by tolerant taxa. Bivalves were the dominant group: *Corbula trigona*, *Anadara senilis* and the oyster *Crassostrea gasar*, all found on the sandy or muddy substrates of the lagoon. Gastropods were mainly represented by Pachymelania and Tympanotonus, colonising muddy bottoms rich in detritus. Finally, the decapod crustacean component included coastal shrimp and crabs such as Penaeus duorarum, *Callinectes latimanus*, *Goniopsis cruentata*, *Clibanarius africanus*, and *Cardiosoma armatum*, which were particularly common in areas with high accumulation of organic matter. The sustained absence of sensitive groups (ephemeroptera, trichoptera, plecoptera) suggests chronic ecological stress linked to oxygen deficits.

Since then, research efforts have focused on ecotoxicology, with particular emphasis on trace metals and their bio-accumulation. Analyses conducted on bivalves and crabs have revealed high levels of Hg, Cd, Cu and Fe, occasionally exceeding WHO thresholds, with marked seasonal variations [45] Similar levels of contamination have also been observed in water and sediments, with peaks during the dry season when the resuspension of fine particles and the intrusion of salt water promote the concentration of dissolved metals [45,46]. Taken together, these results confirm the persistence of chronic chemical stress on macrofauna, but no work has yet established a quantitative link between contamination levels and changes in community composition (specific structure, functional traits) on an interannual scale.

### 3.3.3. Ichthyofauna.

More than 50% of the scientific literature devoted to the Porto-Novo lagoon focuses on ichthyofauna, an interest attested to since the pioneering work of Lemasson (1961) [47], Colleuil and Texier (1987) [34], and Lalèyè (1995) [48], who laid the foundations for knowledge of fish assemblages. Since then, research has been structured around four complementary areas: (i) the diversity and spatio-temporal distribution of fish populations [22,38]; (ii) population biology and functional ecology, particularly growth parameters and length-weight relationships of species [38,49]; (iii) fish health and ecological stress related to heavy metals [40,50,51]; and (iv) human pressures and management responses, from acadja fishing to co-management schemes [33,52].

*3.3.3.1. Species richness and abundance structure*: The first investigations carried out on the Porto-Novo lagoon in 1961 revealed a relatively high overall species richness, with more than 100 species [47]. Three decades later, in the 1990s, various studies reported between 42 and 45 species [53] (Lalèyè & Philippart, 1997). Surveys conducted on the ichthyofauna during the 2015 hydrological cycle (January to December) identified 48 species, 41 genera and 27 families [22]. Another study conducted during the 2015–2016 hydrological cycle (July 2015 to June 2016) identified 54 species divided into 49 genera and 35 families (Vodougnon, et al., 2018). These figures exceed those for the 1990s and reflect both increased sampling efforts and the gradual and more marked penetration of marine species into the lagoon

*3.3.3.2. Seasonal and spatial gradients:* The temporal gradient of the ichthyofauna in the Porto-Novo lagoon is closely linked to salinity dynamics. During periods of high water (salinity <5 ‰), species richness peaks due to the influx of river species and the emergence of juveniles. During low water levels (salinity >5 ‰), the community structure becomes simpler, with biomass concentrated on the tolerant euryhaline species *Sarotherodon melanotheron* and *Ethmalosa fimbriata*. Marine species, such as Caranx hippos and Hyporamphus picarti, only enter the lagoon when salinity exceeds 10–12 ‰ [22,38].

From a spatial perspective, two main areas can be distinguished: the southwest (Djassin-Totché), which is dominated by estuarine species such as Carangidae and Mugilidae; and the northwest, where the influence of the Ouémé River (salinity <5 ‰) increases species richness to up to 15 species per campaign, compared to seven in the southwest [22]. The Jaccard similarity index is 73.5% between the eastern and western sides of the lagoon, including the wetlands connected to the west (Aguégués), which highlights the continuity of habitats along the lagoon#39;s horizontal axis [22]. By contrast, the east-south axis showed only 40% similarity, suggesting an ecological barrier along the vertical axis, likely related to the tide and organic load to the south of the Totchè Channel.

### 3.3.4. Macrophytes and aquatic vegetation.

The first botanical surveys, carried out between 1978 and 1980, revealed that the coastal fringe (riparian zone) was dominated by grasses and helophytes, such as *Paspalum vaginatum*, *Echinochloa pyramidalis*, *Vetiveria nigritana* and *Phragmites australis*. Clumps of Typha and young stands of halophilic ferns, such as *Acrostichum aureum*, also established themselves there. Seasonal floating vegetation, such as *Pistia*

*stratiotes*, *Cyperus papyrus* and *Ipomoea sp.*, was also present [34]. The same authors had already reported the near disappearance of the *Rhizophora racemosa* mangrove, which was reduced to a few relict trees along the Totchè canal.

Water hyacinth (*Eichhornia crassipes*) was absent from the 1980 surveys but became the most widespread floating species on the lagoon by 2005 [33]. Several studies at the turn of the 2010s attributed the rapid expansion of water lilies and water hyacinths to hyper-eutrophication fuelled by excess soluble phosphorus. This proliferation has been described as sometimes blocking navigation and intensifying nocturnal deoxygenation, with the fresh biomass of Eichhornia crassipes estimated at between 12 and 18 kg/m². More recent observations have revealed that the dynamics of floating macrophytes closely follow diffuse nutrient exports from the catchment area, with the organic and labile fraction of phosphorus proving particularly accessible to aquatic flora [37].

**3.3.5.Comparison of the Porto-Novo Lagoon with lagoons in West Africa.** Three urban coastal lagoons in West Africa were analysed in comparison with the Porto-Novo lagoon. These were the Lagos/Lekki complex (Nigeria), the Ebrié lagoon (Côte d#39;Ivoire) and the Sakumo II lagoon (Ghana). All of these systems exhibit five key characteristics in common: (i) a sub-equatorial climate with two rainy seasons, (ii) alternating river-ocean connections via a channel or sandbar, (iii) an average depth of ≤ 3 m, (iv) an urban environment with a population density >3,000 inhabitants per km² in the immediate basin, and (v) combined pressures from nutrients, invasive macrophytes, and domestic and artisanal industrial discharges.

The four countries that are hosting these ecosystems have very different legal frameworks. To date, there has been no formalisation of a framework for the ecological classification of surface waters. However, Ghana is gradually adopting the SDG 6.3.2 classification framework, and Côte d#39;Ivoire is preparing a national BQE reference system. It is evident that all nations have promulgated general legislation pertaining to water quality (see Table 1). However, this legislation remains predominantly focused on sectoral thresholds and does not encompass the incorporation of biological indicators.

From the perspective of regular monitoring, a prerequisite for the establishment of a robust ecological classification framework, the Ébrié lagoon constitutes a distinctive case, with a network of eight monthly stations that has been operational since 2016, including benthic series. Conversely, the Sakumo lagoon is subject to monitoring on a quarterly basis, devoid of a standardised biological component. With regard to the Porto-Novo and Lagos/Lekki lagoons, a variety of measurements from ad hoc studies have been collated. However, the temporal discontinuity of these data represents a significant challenge to the application of a normative ecological classification framework (Table 2).

The four urban lagoons are subject to highly divergent nutrient loads and salinity signatures, which is indicative of strong hydrological dynamics. As demonstrated in Table 3, the levels of nutrients, population density and salinity present within the lagoons vary, thus illustrating the differing degrees of pressure experienced by each. The Lagos/Lekki lagoon complex receives the highest phosphorus flows (approximately 2.7 kg ha⁻¹ year⁻¹), while the Sakumo II lagoon combines an intermediate P load with chronic hypersalinity at the end of the low water period (greater than 50 ‰), linked to the periodic closure of the sandbar. The Porto-Novo and Ébrié lagoons have comparable nutrient pressures, but different salinity signatures and macrophytic cover.

**Table 1. Legal and institutional frameworks relating to some lagoons in West Africa.**

| Country | Main legal reference | Implementing authority* | Reference to ecological classification |
|---------|---------------------|-------------------------|----------------------------------------|
| Benin | Law 2010−44 (water management); draft Water Code 2022 | DSIDH / DG-Eau | No |
| Nigeria | Surface and Groundwater Quality Control Regulations (2011) | NESREA | No |
| Ghana | EPA Water Resources Regs. (2016) + WRC Act 522 (1996) | EPA-Ghana / WRC | SDG 6.3.2 adopted |
| Ivory Coast | Water Code 1998; Water Quality Decree (2020) | ONAD / ANDE | Draft grid in progress |

\* See Supporting Information S1 File, Section S5 for acronym definitions.

**Table 2. Overview of density and nature of monitoring networks of some lagoons in West Africa.**

| Lagoon | Physicochemical (parameters × frequency) | Biota monitored | Hydromorphology |
|---|---|---|---|
| Porto-Novo | 10 parameters / month<br>Very discontinuous series | Phyto; fish (1–2 years) | Spot salinity profiles |
| Lagos/Lekki | 12 parameters / month<br>Discontinuous series covering 2014–2021, 5 stations | Spot of phyto; fish 2019 | Bathymetry |
| Ébrié | 14 parameters / month<br>Discontinuous series covering 2016–2022, 8 stations. | Annual phyto + macro-benthos | Tide gauges, flows measurement |
| Sakumo II | 8 parameters/quarter<br>Discontinuous series covering 2012–2021 | No standardised BQE | Opening / closing monitoring |

**Table 3. Overview of recent pressure indicators and characteristic physicochemical variables in some lagoons in West Africa.**

| Indicator | Porto-Novo | Lagos/Lekki | Ébrié | Sakumo II |
|---|---|---|---|---|
| **Catchment population** (hab/km²) | 3 100<br>[54] | 6 500<br>[55] | 4 800<br>[56] | 4 200<br>[57] |
| **Flux PT**<br>(kg/ha/year) | 1,3<br>[58] | 2,7<br>[59] | 3,5<br>[60] | 1,8<br>[61] |
| $PO_4$-P (mg/L)<br>peak value | 0.30±0.05<br>[32] | 0,85±1,04<br>[62] | 0.55±0.09<br>[60] | 0.57±0.08<br>[61] |
| **Salinity during low water** (‰) | 23±5<br>[32] | 17±4<br>[63] | 3±1<br>[60] | 55±7<br>[61] |
| Chlorophylle **a**<br>(µg/L) | 70±25<br>[32] | 82±35<br>[64] | 45±15<br>[60] | 96±40<br>[61] |
| **Eichhornia** coverage (%) | 35±6<br>[33] | 20±4<br>[63] | ≤ 5<br>[60] | 0<br>([61] |

## 4. Discussion

### 4.1. Completeness of available datasets for applying the SDG 6.3.2 and WFD frameworks to the Porto-Novo lagoon

The UNEP GEMS/Water framework, as defined by the SDG 6.3.2 indicator, and the Water Framework Directive (WFD) have very different assessment requirements. The former is based on monitoring physicochemical variables, whereas the latter requires monitoring of biological indicators, as well as supporting physicochemical variables and hydromorphological descriptors, which are defined in relation to reference conditions. Therefore, the completeness of the available data series for the Porto-Novo lagoon, with a view to applying the UNEP GEMS/Water (SDG 6.3.2) and WFD frameworks, was analysed. This analysis revealed certain gaps that restrict the full application of these two frameworks (see Tables 4 and 5, and Fig 4) (more detail in Supporting Information S1 File, Section S4).

For the SDG 6.3.2 indicator, four of the five 'core' parameter groups (dissolved oxygen, salinity/electrical conductivity, phosphorus and acidification/pH) are monitored. However, the lack of a series for nitrogen and temporal discontinuity (fewer than five consecutive years per station) limit the applicability of the framework to level 1 (Table 4). In the official methodology, it is indeed recommended that the 'nitrogen' group be filled in with the 'total oxidised nitrogen' (TON = $NO_3^-$ + $NO_2^-$) fraction. However, uncertainties are associated with the quantification of $NO_3^-$ in the context of surface waters in southern Benin lagoons because the method generally used (the sodium sacylate assay) is subject

**Table 4. Degree of completeness of Porto-Novo Lagoon datasets against SDG 6.3.2 Level 1 core parameter groups (readiness assessment; period 2006–2022).**

| SDG 6.3.2 parameters | Series ≥ 5 years continuous? | ≥ 10 measurements? | ≥ 3 stations? | Degree of completeness |
|---|---|---|---|---|
| pH | No (discontinuous series, max 3 years) | Yes (≥ 20 measurements, 2006–2022) | Yes | Partially covered |
| Dissolved O₂ | No (discontinuous series, < 3 years) | Yes (≥ 15 measurements, 2006–2022) | Yes | Partially covered |
| Conductivity/salinity | No (series > 5 years, but not continuous) | Yes (≥ 20 measurements, 2006–2022) | Yes | Well covered But without strict continuity |
| PO₄-P | No (episodic data, max 2 years) | Yes (≥ 20 measurements, 2006–2022) | No (1–2 stations) | Very partially covered |
| NH₃-N / NO₂-N | No (very little data available) | No (< 5 measurements, available 2006–2022) | No (1–2 stations) | Very partially covered |

**Table 5. Degree of completeness of available evidence for WFD quality elements in the Porto-Novo Lagoon, and implications for WFD assessment readinessreadiness.**

| WFD layer | WFD element | Available variables / evidence | Frequency / duration | Degree of completeness * |
|---|---|---|---|---|
| Foundational prerequisites | Delineation and typology (water-body boundaries; System A/B typology) | Only general descriptive information on lagoon functioning (freshwater inputs, marine influence/salinity seasonality, connectivity); no formal water-body delineation/coding and no typology assignment | | 1/4 |
| | Reference conditions, EQRs and class boundaries | No defined reference conditions (sites/periods) and no EQR calculation; class boundaries not established | | 0/4 |
| Biological quality elements (BQE) | Phytoplankton diversity and abundance | Chlorophyll-a, phytoplankton richness, abundance reported in multiple studies (heterogeneous methods) | Mostly punctual, short series; limited comparability across years | 2/4 |
| | Macrophytes composition and abundance | Species presence and qualitative coverage reported; limited quantitative mapping | Mostly qualitative and site-specific observations | 1/4 |
| | Macroinverte-brates community structure | Diversity, abundance inventories available (restricted campaigns); indicator potential not standardised | Short campaigns; no long-term standard protocol | 2/4 |
| | Fish fauna diversity and abundance | Strong evidence for fish-related metrics (including repeated and seasonal information), but not under a standard multi-species WFD protocol | Repeated studies exist, but not structured as WFD BQE monitoring | 3/4 |
| Supporting physico-chemical parameters | Nutrients (TP, TN, PO4-P, NO3-N, NO2-N, NH4-N) | Substantial nutrient evidence (TP and PO4 more common than TN species); discontinuities and gaps remain | Mostly discontinuous series; limited multi-year consistency | 3/4 |
| | Oxygen conditions (DO/ BOD where available) | Multiple datasets show hypoxia episodes and seasonal patterns; comparability issues across sources | Discontinuous; not a harmonised monitoring series | 3/4 |
| | Specific pollutants (metals / bioaccumulation) | Metals and bioaccumulation evidence exists (matrices and targets vary) | Sporadic; not compliance-oriented and not harmonised | 3/4 |
| Hydro-morphology (supporting element) | Hydrology, residence time, bathymetry, connectivity, habitat alteration | Hydromorphological information exists (bathymetry, connectivity, uses), but not translated into standardised WFD hydromorpho indicators | Heterogeneous and non-systematic | 2/4 |

\* 0/4 = absent; 1/4 = specific, qualitative data only; 2/4 = existing inventories but no long series and/or no reference conditions; 3/4 = solid quantitative data series but no defined reference conditions; 4/4 = all prerequisites for WFD classification are covered.

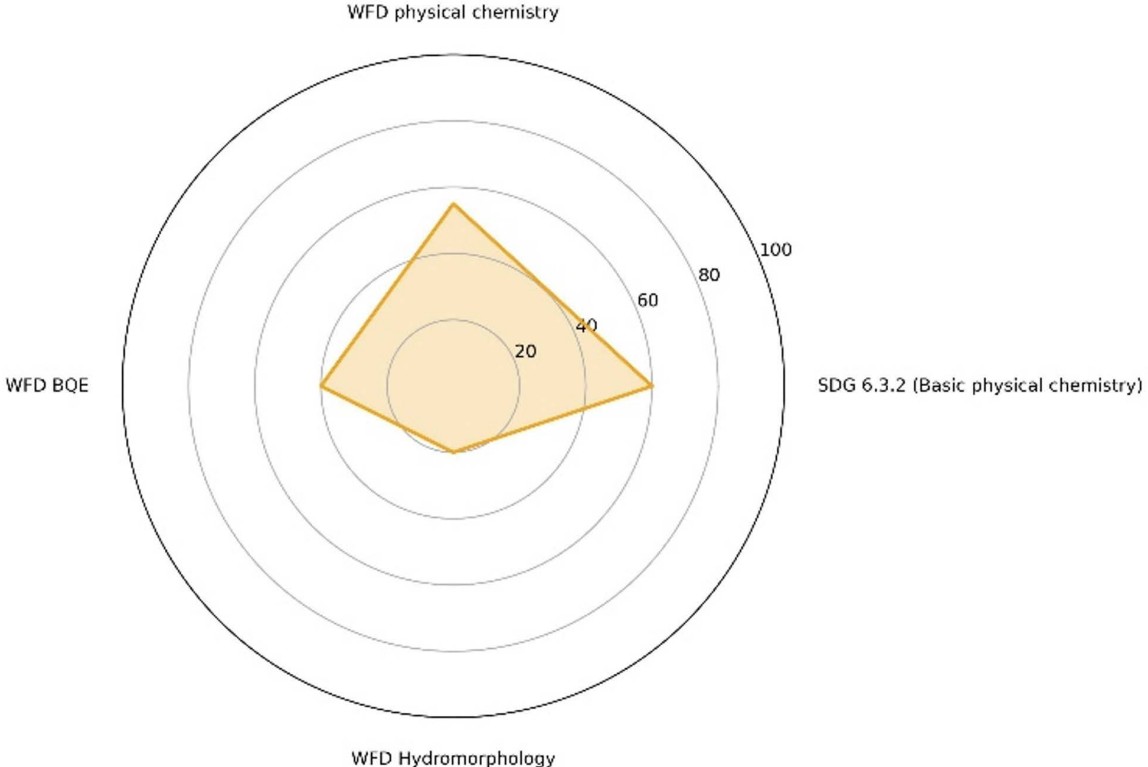

**Fig 4. Degree of completeness (%) of the Porto-Novo lagoon datasets in relation to SDG 6.3.2 and WFD requirements..**

to biases related to salinity, such as optical interference, modified chemical reactions and ionic competition. These biases compromise the reliability and comparability of measurements [65]. Since nitrite generally accounts for less than 1% of TON in surface waters, TON mainly refers to $NO_3^-$. Therefore, it is currently not possible to rigorously meet the requirement of the five explicit parametric groups used to calculate the Porto-Novo Lagoon#39;s compliance rate and binary classification. Also, in terms of the reporting window, the indicator favours data from at least three stations over the last five years. Based on the available data, only one year in this period is acceptable for a level 1 calculation for the Porto-Novo Lagoon. Extending to the period 2006–2022, none of the five parameters cover five consecutive years.

If Level 1 requirements cannot be met for all parameter groups and representative stations, SDG 6.3.2 Level 2 can support context-specific reporting by incorporating additional parameters and complementary information sources to improve coverage; however, it does not replace the need for Level 1 core in situ monitoring for formal reporting. In the Porto-Novo Lagoon, such complementary datasets remain sparse.

Implementing a WFD ecological status assessment implies the ability to produce a traceable and reproducible classification (High/Good/Moderate/Poor/Bad) by following the WFD logic chain: water-body delineation and typology, type-specific reference conditions, standardised biological metrics (BQEs), conversion into EQRs, and the application of class boundaries and aggregation rules. The literature compiled for the Porto-Novo Lagoon provides valuable evidence for several quality elements, particularly supporting physico-chemical variables and selected biological observations. However, data availability does not equate to assessment readiness. In the absence of formal water-body delineation/typology and, critically, type-specific reference conditions enabling EQR computation and class boundary setting, a defensible WFD ecological status classification cannot be produced at this stage (Table 5).

A more detailed appraisal of completeness against WFD quality-element requirements indicates that a decision-grade dataset should cover: (i) biological quality elements (BQEs); (ii) supporting physico-chemical elements; and (iii) hydro-morphological descriptors. These components underpin ecological status determination, while chemical status relies on compliance with environmental quality standards for priority substances. The two are combined to derive an overall status, driven by the more stringent of the two classifications.

In the Porto-Novo Lagoon, none of the above-mentioned BQEs are supported by consistent spatially representative and temporally comparable time series, and hydromorphological descriptors remain sporadic and non-standardised (Table 5). In addition, the biological evidence base is strongly unbalanced across BQEs. Macroinvertebrate datasets remain sparse and non-standardised, and zooplankton is only intermittently documented, which further constrains any assessment-grade WFD-type interpretation. This prevents any robust WFD-type classification. In addition, chemical status cannot currently be evaluated because monitoring is not structured around WFD-relevant priority substances and compliance objectives. Given the local context of intensive agriculture and limited domestic/industrial wastewater treatment, this gap is particularly critical for compounds such as pesticides and metals, which are only documented in the literature in a fragmented manner. Consequently, the WFD component is presented as an exploratory and diagnostic readiness (gap) analysis rather than an operational WFD assessment, given the absence of type-specific reference conditions and continuous BQE time series; a defensible WFD ecological status classification cannot be produced at this stage.

The overall degree of incompleteness for SDG 6.3.2 is 60%, for the WFD#39;s Biological Quality Elements it is 40%, for the WFD#39;s physico-chemistry it is 55%, and for the WFD#39;s hydromorphology it is only 20% (Fig 4). These values provide an ordinal gap visualisation of evidence coverage and should be interpreted as readiness constraints, not as a validated quantitative metric or a compliance-based classification outcome. In particular, BQE completeness reflects evidence structure (time series, protocol consistency) rather than ecological importance; qualitative biological information informs ecological context but is not assessment-ready. Overall, the gaps identified remain a major technical barrier to any robust, policy-relevant classification and highlight the well-known trade-off between frameworks: SDG 6.3.2 is more rapidly mobilised but less sensitive to structural alterations, whereas the WFD is more integrative but requires richer datasets and substantial calibration effort.

## 4.2. Readiness for SDG 6.3.2 reporting and WFD-oriented gap analysis

### 4.2.1. SDG 6.3.2: Readiness for Level 1 reporting and role of context-adapted complements. SDG 6.3.2 reporting can rely on multi-year assessment windows, which can be useful where annual series are incomplete. However, formal UNEP Level 1 reporting requires consistent in situ monitoring of the five core parameter groups and the application of nationally defined target values to classify representative water bodies as "good/not good". Given the heterogeneity of the available literature sources and the gaps identified for the Porto-Novo Lagoon particularly for the nitrogen group and the lack of continuous multi-year station series the short-term perspective is best framed as readiness building (strengthening consistent in situ series), rather than immediate formal reporting. Complementary, context-adapted data sources (often discussed under Level 2 approaches, e.g., quality-assured citizen measurements or Earth observation for optically detectable parameters) may help address spatial/temporal gaps and support interpretation, but should be presented as complements rather than substitutes for Level 1 requirements.

### 4.2.2. WFD: Prerequisites and roadmap. The implementation of a classification system in accordance with the WFD necessitates the characterisation and delimitation of surface water bodies within the river basin district. This includes the typology and establishment of reference conditions for transitional waters. These contents are structured and reported at the level of surface water bodies, encompassing characterisation, pressures/impacts, ecological status, chemical status, exemptions, and mixing zones. The absence of this architecture renders the assessment and comparison with ecological standards both unfeasible and unauditable.

With regard to the Porto-Novo lagoon, the short-term readiness is limited by the current absence of three elements. Firstly, there is an absence of interannual and intercalibrated series for Environmental Quality Standards (EQS) adapted

to tropical lagoon environments. Secondly, there is an absence of a systematic hydromorphological dataset. Thirdly, there is an absence of monitoring of priority substances enabling chemical status assessment. In the medium term, partial readiness can be considered as realistic following the following sequence of events: (1) the delimitation/coding of surface water bodies and selection of the relevant "transitional water" type; (2) the pragmatic definition of reference conditions (low-impact sites and/or regional nesting); (3) the design of an integrated network (BQE + physico-chemistry + hydromorphology) with seasonal time steps; (4) the implementation of monitoring of priority substances (water and biota) and compliance thresholds; (5) the documented and traceable aggregation protocol ("one-out, all-out"). These steps are designed to align the database with reporting requirements and ensure inter-annual and inter-site comparability.

### 4.3. Institutional and operational implications of applying a classification framework in West Africa

A thorough analysis of the implementation of classification frameworks (SDG 6.3.2 in the short term; WFD in the medium/long term) reveals that success is contingent on the institutional architecture and the regularity of monitoring, rather than on the selection of metrics alone. For Porto-Novo, four principles emerge, informed by the trajectories observed in Ébrié, Sakumo II and the Lagos/Lekki complex: governance, regulatory progress, financial sustainability and community involvement.

In terms of governance, stable management and operational continuity are the key determinants. The example of Ébrié demonstrates that a clearly identified operator, with a technical mandate and a monthly schedule, produces series that are compatible with a cumulative and robust ranking [60,66]. Conversely, the institutional fragmentation documented around Lagos/Lekki and, to a certain extent, in Porto-Novo, where hydrology falls under the remit of water authorities, chemistry is the responsibility of university laboratories and macrophyte management is the responsibility of municipalities, increases the risk of methodological "silos" and temporal discontinuities. The challenge confronting Porto-Novo is to establish a formalised scientific coordination and data management system. One potential approach involves the establishment of a "lagoon authority" arrangement, involving the environmental authority, the DSIDH and the riverside municipalities. This system would be based on a common reference framework, encompassing stations, methods and metadata. The objective of this initiative is to transform ad hoc campaigns into a series of utilitarian chronicles.

From a regulatory perspective, a gradual approach is the most pragmatic. In Ghana, the adoption of the SDG 6.3.2 matrix by the EPA and the WRC, with a national dashboard since 2023 providing information primarily on the five mandatory variables, provides a credible classification basis before any biological ambitions. In Côte d#39;Ivoire, a "WFD-light" reference framework focused on adapted EQS is in preparation; in Benin and Nigeria, the texts remain focused on sectoral thresholds for the time being. For Porto-Novo, the pivotal step is to make the SDG indicator immediately calculable by supplementing the physico-chemical core with a nitrogen fraction analysed robustly in brackish environments (e.g., In the absence of TON, alternative options such as $NH_4^+$ should be considered, with explicit targets in place. Subsequent to this, WFD pilots can be initiated, utilising fundamental BQE and basic hydromorphological descriptors. These pilots will draw upon regional experiences, including the tropical adaptation of a macrobenthic index inspired by the M-AMBI, which was previously tested in Lagos [63]

The issue of funding is less about initial investment and more about ensuring operational sustainability. In Ébrié, a sanitation fee (USD 0.08–0.12/m³) contributes to financing laboratory and operating costs (Awuah et al., 2021). In Sakumo II, the financing of multi-parameter instrumentation is ensured by budget lines from Ramsar and the EPA [61]. In Porto-Novo, the reliance on ad hoc research grants has been identified as a key factor contributing to the weakening of the network and the compromise of inter-annual comparability. A credible option, based on regional lessons learned, is to back monitoring with a parafiscal mechanism combining, for example, a tax on sand extraction, navigation fees and effluent charges, guaranteeing recurring items (consumables, calibration, maintenance, analyses) for a smaller but regular network.

The involvement of local residents represents a dual function, serving as both a social and a scientific lever. In the Lagos/Lekki area, the integration of co-design mechanisms for stations and the supervision of citizen observations has enhanced the spatial granularity and relevance of diagnoses. In Ghana, the implementation of quarterly community

mapping of water hyacinths has led to a substantial reduction in vegetation cover within a brief period [67]. In the context of Porto-Novo, where ecological concerns are intricately interwoven with community dynamics, the implementation of such a scheme has the potential to enhance the detection of events (e.g., blooms, fish mortality, salinity anomalies), stabilise measurement sites, and increase the credibility of results. This approach would also necessitate coordination with the application of coastal easements as stipulated by national regulations.

## 5. General conclusion

The study under scrutiny highlights a dual paradox inherent in the Porto-Novo lagoon. Firstly, the accumulation of a mass of scattered data over a period of four decades is clearly documented. This data is indicative of the evolution of a shallow system subject to marked seasonal salinity forcing and persistent eutrophication. Secondly, the information remains too incomplete and heterogeneous to allow for a comprehensive ecological classification in the sense of a rigorous ecological classification. However, the minimum requirements of SDG 6.3.2 appear achievable in the short term: integrating reduced nitrogen into existing monthly campaigns will provide the required set of variables in less than five years, thereby allowing the category to be removed from the "not assessed" list. In the longer term, the transposition of a WFD light to tropical lagoons necessitates the coordinated development of tropical indices and the implementation of a simplified hydromorphological module.

The readiness of this gradual trajectory is contingent upon three conditions: the establishment of a unified scientific governance framework, the creation of sustainable funding mechanisms, and the active involvement of grassroots communities. When considered as a whole, these elements suggest a trajectory in which Porto-Novo first consolidates a core of regular and traceable monitoring calibrated to SDG 6.3.2 (with appropriate management of analytes sensitive to brackish environments), then launches targeted WFD pilots as soon as the biological and hydromorphological series become consistent. The Porto-Novo classification will be able to move beyond a declarative exercise to become part of a cumulative, reproducible and scientifically robust system if it draws inspiration from the operational stability of Ébrié, avoids the pitfalls of frequency and standardisation highlighted in Sakumo II, and addresses from the outset the multi-stakeholder coordination and data interoperability revealed by the Lagos/Lekki case.

## Supporting information

**S1 File. Supplementary material for the ecological classification review of the Porto-Novo Lagoon.** This file contains additional methodological notes, framework requirements, calculation procedures, completeness-score explanations, and acronym definitions.
(DOCX)

**S2 Checklist. PRISMA-ScR checklist for this scoping review.**
(DOCX)

## Author contributions

**Conceptualization:** Metogbe Belfrid Djihouessi, Gildas Djidohokpin.

**Data curation:** Gildas Djidohokpin, Romaric Christian Marc Hekpazo, Akilou Amadou Socohou, Lewis Zounon.

**Formal analysis:** Metogbe Belfrid Djihouessi, Gildas Djidohokpin, Romaric Christian Marc Hekpazo, Akilou Amadou Socohou.

**Investigation:** Lewis Zounon.

**Methodology:** Metogbe Belfrid Djihouessi, Akilou Amadou Socohou.

**Supervision:** Zacharie Sohou, Abou Youssouf, Martin Aina.

 

**Writing – original draft:** Metogbe Belfrid Djihouessi.

**Writing – review & editing:** Metogbe Belfrid Djihouessi, Gildas Djidohokpin, Romaric Christian Marc Hekpazo, Akilou Amadou Socohou, Zacharie Sohou, Abou Youssouf, Martin Aina.

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
