## [Decision Letter · Decision Letter 0]

14 Jan 2026

Dear Dr. Djihouessi,

Thank you for submitting your manuscript to PLOS ONE. After careful consideration, we feel that it has merit but does not fully meet PLOS ONE’s publication criteria as it currently stands. Therefore, we invite you to submit a revised version of the manuscript that addresses the points raised during the review process.

We look forward to receiving your revised manuscript.

Kind regards,

Mehrnoush Aminisarteshnizi, Ph.D.

Academic Editor

PLOS One

Journal Requirements:

3. We note that Figure 2 in your submission contain [map/satellite] images which may be copyrighted. All PLOS content is published under the Creative Commons Attribution License (CC BY 4.0), which means that the manuscript, images, and Supporting Information files will be freely available online, and any third party is permitted to access, download, copy, distribute, and use these materials in any way, even commercially, with proper attribution. For these reasons, we cannot publish previously copyrighted maps or satellite images created using proprietary data, such as Google software (Google Maps, Street View, and Earth). For more information, see our copyright guidelines: http://journals.plos.org/plosone/s/licenses-and-copyright.

1. You may seek permission from the original copyright holder of Figure 2 to publish the content specifically under the CC BY 4.0 license.

**Additional Editor Comments:**

Dear Authors,

Thank you for submitting your manuscript entitled

*“Ecological patterns in the Porto-Novo Lagoon (Benin, West Africa): a review with implications for SDG 6.3.2 and EU-WFD ecological status assessment”* to *PLOS ONE*.

The manuscript has now been evaluated by two independent reviewers. Both reviewers recognize the relevance, scope, and policy significance of the study, as well as the value of compiling long-term ecological information for a data-limited tropical lagoon. However, they also identified several issues that require clarification before the manuscript can be considered for publication.

Based on the reviewers’ reports and my own assessment, I am pleased to inform you that the manuscript is invited for **Major Revision**.

In revising your manuscript, please pay particular attention to the following points raised by the reviewers:

1. Clarify the methodological basis for assigning data “completeness” and “maturity” scores, including thresholds and decision rules.

2. Clearly state that the application of the EU Water Framework Directive is diagnostic and exploratory, rather than a formal operational classification.

3. Improve the reporting and integration of the statistical analyses mentioned (or alternatively reframe the analysis as primarily descriptive, if appropriate).

4. More explicitly acknowledge the uneven availability of Biological Quality Element data and its implications for WFD applicability.

Please provide a detailed, point-by-point response to each reviewer comment, indicating how and where changes have been made in the revised manuscript.

We look forward to receiving your revised submission.

Sincerely,

Reviewers' comments:

Reviewer's Responses to Questions

**Comments to the Author**

1. Is the manuscript technically sound, and do the data support the conclusions?

Reviewer #1: Yes

Reviewer #2: Partly

2. Has the statistical analysis been performed appropriately and rigorously?

Reviewer #1: I Don't Know

Reviewer #2: Yes

3. Have the authors made all data underlying the findings in their manuscript fully available?

Reviewer #1: Yes

Reviewer #2: Yes

4. Is the manuscript presented in an intelligible fashion and written in standard English?

Reviewer #1: Yes

Reviewer #2: Yes

Reviewer #1: The paper entitled “Ecological patterns in the Porto-Novo Lagoon (Benin, West Africa): a review with implications for SDG 6.3.2 and EU-WFD ecological status assessment” was revised.

The study addresses an important and timely topic, and the compilation of long-term environmental data from a data-limited tropical lagoon is potentially valuable. However, after careful evaluation, the manuscript requires significant revision before it can be considered further.

The primary concern is that the manuscript overextends its conclusions relative to the analytical framework and data structure used. While the review assembles a substantial body of historical information, it does not yet provide a sufficiently rigorous or transparent methodological basis to support claims regarding the applicability of the EU Water Framework Directive (WFD) or operational readiness for SDG 6.3.2 reporting.

In particular:

1. Use of the WFD framework

The manuscript discusses WFD applicability without the foundational elements required for such an assessment (e.g. water body typology, reference conditions, ecological quality ratios). As a result, statements implying partial application or feasibility are not adequately supported. The WFD component should be reframed as a conceptual gap analysis, not an assessment.

2. Scoping review methodology

Although PRISMA-ScR is cited, key methodological details (search strategy, screening procedure, data extraction framework, and quality control) are insufficiently documented. This limits reproducibility and weakens the evidentiary basis of the synthesis.

3. Completeness scoring system

The scoring and aggregation approach used to evaluate data completeness is author-defined and lacks clear justification or validation. The interpretation of these scores should be substantially tempered, and the limitations of this approach made explicit.

4. SDG 6.3.2 interpretation

The manuscript does not clearly distinguish between formal UNEP reporting requirements and context-adapted proxy approaches. Claims regarding short-term feasibility should be moderated and aligned more explicitly with official SDG guidance.

5. Biological quality elements

The treatment of biological data is uneven, and qualitative information is in some cases presented as assessment-ready. Greater caution is required when linking ecological knowledge to policy-relevant evaluation.

Overall, the manuscript would benefit from tightening its scope, moderating claims, and improving methodological clarity. Substantial restructuring and clarification are needed to ensure that conclusions are firmly supported by the review design and available data.

Reviewer #2: This manuscript provides a thorough scoping review of ecological data from the Porto-Novo Lagoon and evaluates their suitability for application of the UNEP SDG 6.3.2 indicator and the EU Water Framework Directive (WFD). The topic is relevant, timely, and well aligned with the scope of PLOS ONE. The regional comparison and explicit science–policy linkage are clear strengths.

Major Comments

Methodological transparency

The criteria used to assign “completeness” and “maturity” scores are not fully explicit. A clearer description of thresholds and decision rules is needed to ensure reproducibility.

Interpretation of WFD applicability

The manuscript should more clearly state that the WFD assessment is exploratory/diagnostic rather than operational, given the lack of reference conditions and continuous BQE datasets.

Statistical analyses

The use of Mann–Kendall and ANOVA is mentioned, but results are not clearly reported. Either provide full statistical outputs or reframe the study as primarily descriptive.

Biological data gaps

Strong imbalance exists among Biological Quality Elements, with limited data for macroinvertebrates and zooplankton. This limitation should be more explicitly reflected in the discussion of WFD feasibility.

Minor Comments

The abstract could be slightly shortened.

Some technical terms require clearer definition at first use.

Minor language editing is recommended for clarity and flow.

.

Reviewer #1: No

Reviewer #2: No

---

## [Author Response · Author response to Decision Letter 1]

5 Mar 2026

Response to reviewers comments

PONE-D-25-55877

Ecological patterns in the Porto-Novo Lagoon (Benin, West Africa): a review with implications for SDG 6.3.2 and EU-WFD ecological status assessment.

Response to editor

We would like to thank the Editor and the two reviewers for their constructive comments. We have revised the manuscript accordingly to improve methodological transparency and clarify the scope as a readiness and gap analysis (WFD) and SDG 6.3.2 Level 1 readiness evidence mapping. We have also tempered interpretations where needed and strengthened the treatment of biological evidence and limitations. To ensure full consistency with this revised scope, the manuscript title has been slightly adjusted to: “Ecological patterns in the Porto-Novo Lagoon (Benin, West Africa): a review with implications for SDG 6.3.2 and EU-WFD readiness toward ecological status classification.” All changes are highlighted in the revised manuscript (in yellow for changes related to Reviewer 1's comments and in green for changes related to Reviewer 2's comments) and are detailed in a point-by-point response below..

About the figure 2

Thank you for your comments regarding the copyright status of Figure 2.

We would like to assure you that Figure 2 has been corrected and does not contain any proprietary or previously copyrighted cartographic elements (e.g., Google Maps, Google Earth, or other restricted sources). This map was derived exclusively from Sentinel-2 Level 2A images, imagery acquired on 20 Dec 2024 and downloaded on 19 Feb 2026, obtained from the Copernicus Open Access Hub under the Copernicus data and information open access policy (European Union, Copernicus program).

Sentinel-2 data are freely available for use and redistribution and are compatible with publication under a Creative Commons Attribution (CC BY 4.0) license. The figure was processed and produced independently by the authors using the free software QGIS. The map is therefore an original work created from freely available satellite data.

To ensure complete transparency, we have updated the figure legend to clearly indicate the data source and licensing information.

Response to Reviewer #1 comments:

1. Use of the WFD framework

The manuscript discusses WFD applicability without the foundational elements required for such an assessment (e.g. water body typology, reference conditions, ecological quality ratios). As a result, statements implying partial application or feasibility are not adequately supported. The WFD component should be reframed as a conceptual gap analysis, not an assessment.

Thank you for this important clarification. We agree that a defensible WFD ecological status assessment requires foundational building blocks that are not currently in place for the Porto-Novo Lagoon. These include formal water-body delineation and typology, type-specific reference conditions and the EQR-based framework resulting from these, with class boundaries and aggregation rules. In the revised manuscript, we therefore reframe the WFD component as an explicit conceptual readiness/gap analysis rather than an assessment. To reflect this change consistently, the manuscript title has been adjusted accordingly by replacing “EU-WFD ecological status assessment” with “EU-WFD readiness toward ecological status classification.”

Specifically, we:

(i) revise the Methods section to clarify that the WFD-related scoring is a data readiness score (0–4), which is used to benchmark the availability of evidence against WFD prerequisites. This score is not an EQR and does not produce WFD status classes.

(ii) revise the WFD section headings and narrative to avoid wording that suggests the partial application or feasibility of a WFD assessment.

(iii) strengthen Table 4.2 by adding the missing foundational prerequisites (water body delineation and typology, reference conditions and ecological quality ratios), and by inserting an explicit note stating that the availability of quality element evidence does not equate to WFD assessment readiness. These revisions ensure that the WFD discussion is aligned with what the compiled literature can legitimately support, while also providing a transparent roadmap of what is required for any future WFD-inspired classification effort.

2. Scoping review methodology

Although PRISMA-ScR is cited, key methodological details (search strategy, screening procedure, data extraction framework, and quality control) are insufficiently documented. This limits reproducibility and weakens the evidentiary basis of the synthesis.

Thank you for this comment. In the revised manuscript, we have substantially strengthened the methodology by adding the missing operational details.

First, we now document the search strategy by specifying the information sources (Scopus, Web of Science, AJOL, HAL, IRD Horizon, Google Scholar as a search engine, and local academic libraries), the time window (1970–2024), languages (French/English), the date of the last search (September 2025), and the core Boolean query, including how it was implemented across sources (TITLE-ABSTRACT-KEYWORDS in Scopus; TOPIC in Web of Science; adapted fields in AJOL/repositories; free-text in Google Scholar).

Second, we clarify the screening procedure, including deduplication in Mendeley, a two-step screening workflow (title/abstract screening followed by full-text assessment) consistent with the PRISMA flow diagram, and explicit inclusion/exclusion criteria, including duplicate dataset handling.

Third, we clarify the data extraction framework by describing the use of a predefined 22-field charting mask, the application of controlled vocabularies for variables and matrices, and the systematic coding of missing information as “not reported” to ensure consistent data structure across heterogeneous sources.

Finally, we document quality control measures applied to the charted dataset, including unit consistency checks and harmonisation/flagging, cross-checking duplicates to avoid double counting, and internal consistency checks to detect obvious transcription anomalies. These revisions improve transparency and reproducibility and strengthen the evidentiary basis of the synthesis...

3. Completeness scoring system

The scoring and aggregation approach used to evaluate data completeness is author-defined and lacks clear justification or validation. The interpretation of these scores should be substantially tempered, and the limitations of this approach made explicit.

Thank you for this comment. In the revised manuscript, we have therefore tempered the interpretation and made the limitations explicit.

Specifically, we revised Section 2.4 to :

(i) clearly state the purpose of the 0–4 score as an ordinal data-readiness / evidence-coverage heuristic used for structured gap mapping;

(ii) define the score levels transparently (0/4 = absent; 1/4 = qualitative/site-specific evidence only; 2/4 = inventories without consistent time series and/or without reference context; 3/4 = repeated quantitative series but no defined reference conditions; 4/4 = prerequisites for traceable WFD classification met); and

(iii) explicitly acknowledge that the scoring is not validated, is not an EQR, and does not provide WFD status classes.

We also added a limitations statement noting that the scoring does not weight spatial representativeness, analytical comparability across studies, or quality control heterogeneity, and that results may be sensitive to the chosen thresholds for defining time-series length. Accordingly, we now present score differences as indicative gap patterns rather than quantitative distances to compliance or ecological status (Table 4.2; Figure 4).

4. SDG 6.3.2 interpretation

The manuscript does not clearly distinguish between formal UNEP reporting requirements and context-adapted proxy approaches. Claims regarding short-term feasibility should be moderated and aligned more explicitly with official SDG guidance.

Thank you for this important clarification. The manuscript has been revised to clearly distinguish between formal UNEP SDG 6.3.2 reporting requirements (Level 1) and context-adapted proxy/evidence-mapping approaches derived from heterogeneous literature.

Concretely, the Methods now state that formal Level 1 reporting is based on in situ monitoring of the five core parameter groups and the application of national target values to classify representative water bodies as “good/not good”, whereas the present study performs a readiness/evidence-coverage audit using the variables most consistently reported in the literature. The Results/Discussion have been revised accordingly: statements implying short-term “feasibility” of official reporting have been tempered, and Level 2-type additions (e.g., quality-assured citizen measurements and Earth observation for optically detectable parameters) are now presented explicitly as complements to support gap filling and contextual interpretation, not as substitutes for Level 1 requirements. Overall, SDG-related claims are now aligned with official SDG guidance and framed as a staged pathway toward reporting readiness rather than immediate formal compliance-based reporting.

5. Biological quality elements

The treatment of biological data is uneven, and qualitative information is in some cases presented as assessment-ready. Greater caution is required when linking ecological knowledge to policy-relevant evaluation.

Thank you for the comment. In the revised manuscript, we strengthened caution when linking ecological knowledge to policy-relevant evaluation.

Specifically, the Results in Biota section now explicitly state that qualitative biological observations are synthesised for ecological context and pressure interpretation only, and not as assessment-grade inputs. Wording that could be interpreted as implying readiness (e.g., “healthy”/“confirming” statements) has been moderated. In the Discussion around the completeness/readiness synthesis in Table 4.2 and Figure 4, interpretation has been tempered to clarify that BQE completeness reflects evidence structure rather than ecological importance, and that uneven maturity across BQEs limits policy-relevant evaluation. Overall, biological findings are now used to prioritise candidate indicators and monitoring needs, rather than to infer assessment-ready status classes.

Response to Reviewer #2 comments:

Major Comments

1. Methodological transparency

The criteria used to assign “completeness” and “maturity” scores are not fully explicit. A clearer description of thresholds and decision rules is needed to ensure reproducibility.

Thank you for this comment.

The manuscript has been revised to make the completeness (“degree of completeness/data-readiness”) scoring fully explicit and reproducible. In addition to defining the score levels (0/4 to 4/4) and the SDG readiness threshold (≥10 observations over five years), the revised text now specifies the key decision rules used to assign scores across heterogeneous sources (best available evidence meeting the criteria; duplicate reporting cross-checked to avoid double counting, retaining the most complete primary record). The aggregation rule used to derive the “degree of incompleteness” summaries (Figure 4) is also stated explicitly, including equal weighting across parameters within each framework block and the formula used for computation (with max score as the maximum possible score and observed score as the assigned score). These additions address transparency and ensure that the scoring and aggregation can be replicated.

2. Interpretation of WFD applicability

The manuscript should more clearly state that the WFD assessment is exploratory/diagnostic rather than operational, given the lack of reference conditions and continuous BQE datasets.

Thank you for this comment.

The manuscript has been revised to state more explicitly that the WFD component is exploratory and diagnostic rather than operational. In the Discussion section introducing the WFD analysis (before Table 4.2), the text now clarifies that—because type-specific reference conditions are not established and continuous, decision-grade BQE time series are not available for the Porto-Novo Lagoon—the WFD discussion is presented as a readiness (gap) analysis and does not produce an EQR-based status class or an operational WFD ecological status assessment

3. Statistical analyses

The use of Mann–Kendall and ANOVA is mentioned, but results are not clearly reported. Either provide full statistical outputs or reframe the study as primarily descriptive.

Thank you for this comment.

We agree that mentioning inferential tests without reporting full statistical outputs reduces clarity. The manuscript has therefore been revised to remove the reference to Mann–Kendall and repeated-measures ANOVA, and to frame the study explicitly as a primarily descriptive scoping synthesis. Temporal patterns are now discussed only as descriptive evidence drawn from the compiled literature, consistent with the heterogeneous, non-standardised and often discontinuous nature of the underlying datasets.

4. Biological data gaps

Strong imbalance exists among Biological Quality Elements, with limited data for macroinvertebrates and zooplankton. This limitation should be more explicitly reflected in the discussion of WFD feasibility.

Thank you for this comment.

We agree that the biological evidence base is uneven across Biological Quality Elements. The manuscript has been revised to reflect this more explicitly in the discussion of WFD applicability: in addition to the absence of type-specific reference conditions and continuous BQE time series, macroinvertebrate datasets remain sparse and non-standardised, and zooplankton is only intermittently documented. We now state directly (in the WFD discussion introducing Table 4.2) that this imbalance further constrains any assessment-grade WFD-type interpretation and reinforces that the WFD component remains a diagnostic readiness (gap) analysis rather than an operational assessment.

Minor Comments

5. The abstract could be slightly shortened.

Thank you for this suggestion. The abstract has been updated to reflect all the changes made to the manuscript.

6. Some technical terms require clearer definition at first use.

Thank you for this comment. The manuscript has been revised to improve clarity by defining technical terms at first use. Specifically, acronyms and framework-specific concepts (BQEs, EQR, EQS) are now expanded and briefly defined when they first appear, ensuring that subsequent uses are unambiguous.

7. Minor language editing is recommended for clarity and flow.

Thank you for this suggestion. The manuscript has been carefully revised for clarity and flow, with minor language editing applied throughout. We corrected grammar and punctuation, improved sentence structure, reduced repetition, harmonised terminology across sections (SDG 6.3.2 and WFD), and adjusted wording to ensure consistent and precise use of technical terms. These edits were made without changing the scientific content

---

## [Decision Letter · Decision Letter 1]

12 Apr 2026

Ecological patterns in the Porto-Novo Lagoon (Benin, West Africa): a review with implications for SDG 6.3.2 and EU-WFD readiness toward ecological status classification.

PONE-D-25-55877R1

Dear Dr. Djihouessi,

We’re pleased to inform you that your manuscript has been judged scientifically suitable for publication and will be formally accepted for publication once it meets all outstanding technical requirements.

Kind regards,

Mehrnoush Aminisarteshnizi, Ph.D.

Academic Editor

PLOS One

Additional Editor Comments (optional):

Reviewers' comments:

Reviewer's Responses to Questions

**Comments to the Author**

Reviewer #1: All comments have been addressed

Reviewer #2: All comments have been addressed

2. Is the manuscript technically sound, and do the data support the conclusions?

Reviewer #1: Yes

Reviewer #2: Yes

3. Has the statistical analysis been performed appropriately and rigorously?

Reviewer #1: Yes

Reviewer #2: Yes

4. Have the authors made all data underlying the findings in their manuscript fully available?

Reviewer #1: Yes

Reviewer #2: Yes

5. Is the manuscript presented in an intelligible fashion and written in standard English?

Reviewer #1: Yes

Reviewer #2: Yes

Reviewer #1: The authors have submitted a revised version of the manuscript and have provided detailed responses to all the reviewers’ comments.

Reviewer #2: The authors have addressed most of the reviewers’ concerns in a satisfactory and professional manner. The manuscript has significantly improved, particularly in terms of methodological transparency, clarification of scope (WFD as a readiness analysis), and moderation of interpretations.

However, a few minor issues remain, including residual formatting errors, the need to further clarify the novelty and contribution of the study, and minor improvements in data availability and ecological interpretation.

.

Reviewer #1: No

Reviewer #2: No

---

## [Editor Report · Acceptance letter]

PONE-D-25-55877R1

PLOS One

Dear Dr. Djihouessi,

I'm pleased to inform you that your manuscript has been deemed suitable for publication in PLOS One. Congratulations! Your manuscript is now being handed over to our production team.

Kind regards,

on behalf of

Dr. Mehrnoush Aminisarteshnizi

Academic Editor

PLOS One